# Impact of *Mycobacterium tuberculosis* Infection on Human B Cell Compartment and Antibody Responses

**DOI:** 10.3390/cells11182906

**Published:** 2022-09-17

**Authors:** Marco P. La Manna, Mojtaba Shekarkar-Azgomi, Giusto D. Badami, Bartolo Tamburini, Costanza Dieli, Paola Di Carlo, Teresa Fasciana, Vito Marcianò, Bruna Lo Sasso, Rosaria V. Giglio, Anna Giammanco, Marcello Ciaccio, Francesco Dieli, Nadia Caccamo

**Affiliations:** 1Central Laboratory of Advanced Diagnosis and Biomedical Research (CLADIBIOR), Azienda Ospedaliera Universitaria Policlinico (A.O.U.P.) Paolo Giaccone, University of Palermo, 90127 Palermo, Italy; 2Department of Biomedicine, Neurosciences and Advanced Diagnostic (Bi.N.D.), University of Palermo, 90127 Palermo, Italy; 3Department of Sciences for Health Promotion, Mother & Child Care, University of Palermo, 90127 Palermo, Italy; 4Division of Infectious Diseases, Azienda Ospedaliera Universitaria Policlinico (A.O.U.P.) Paolo Giaccone, University of Palermo, 90127 Palermo, Italy; 5Department of Laboratory Medicine, Azienda Ospedaliera Universitaria Policlinico (A.O.U.P.) Paolo Giaccone, 90127 Palermo, Italy

**Keywords:** tuberculosis, B lymphocytes, memory subsets, humoral immunity, pre-existing antibodies

## Abstract

Tuberculosis (TB) remains one of the most important health challenges worldwide. Control of the TB epidemic has not yet been achieved because of the lack of an effective vaccine and rapid and sensitive diagnostic approaches, as well as the emergence of drug-resistant forms of *M. tuberculosis*. Cellular immunity has a pivotal role against *M. tuberculosis* infection, but the role of humoral immunity is still controversial. We analyzed the frequency, absolute counts, and phenotypic and functional subsets of B lymphocytes in the peripheral blood of patients with active TB and subjects with latent infection compared to healthy donors. Moreover, we analyzed serum levels of total Ig and their IgA, IgM, and IgG isotypes and the titers of preexisting antibodies against a pool of common viral pathogens. FlowCT and unsupervised clusterization analysis show that patients with active TB and LTBI subjects have modest non-significant reduction in the numbers of circulating B lymphocytes as compared to healthy donors. Moreover, LTBI subjects had high percentages of atypical B cell population and lower percentages of naive and switched memory B cells. These findings were supported by gene expression and GSEA analysis. Moreover, there were no differences between active TB patients, LTBI subjects and HD, either in serum levels of total Ig isotypes or in preexisting IgG antibody titers, to ten different antigens from eight common pathogenic viruses, clearly demonstrating that either active or latent *M. tuberculosis* infection preserves the antibody production capacity of long-lived plasma cells. Thus, our results agree with previous studies reporting unaltered B cell frequencies in the blood of active TB patients and LTBI individuals as compared to healthy controls.

## 1. Introduction

Tuberculosis (TB) is a severe chronic infectious and inflammatory disease caused by the bacillus *Mycobacterium* (*M.*) *tuberculosis*. To date, it remains one of the most significant causes of mortality throughout the world, and the current situation shows a worsening trend due to the emergence of new antibiotic-resistant strains and the spread of immunosuppressive diseases and therapies that compromise the immune system. Furthermore, completely effective vaccines against the disease are not yet available, as the Calmette and Guèrin vaccine (BCG) offers only partial protection [1].

The World Health Organization (WHO) has estimated that in 2020, 9.9 million people in the world fell ill with TB, equivalent to 127 cases per 100,000 inhabitants. Between 2019 and 2020, deaths increased from 1.2 million to 1.3 million among HIV-negative people and from 209,000 to 214,000 among HIV-positive people. This is the first annual increase in the number of people dying from tuberculosis since 2005, also partially driven by disruptions to supply and access to essential tuberculosis diagnosis and treatment services during the COVID-19 pandemic, leading to an 18% reduction in the number of newly diagnosed cases. In fact, in 2020, tuberculosis ranked second as a cause of death by a single infectious agent after SARS-CoV-2, a record it held in 2019 [2].

Protective immune responses against *M. tuberculosis* remain not clearly defined. The current paradigm underscores the important role for T-cell mediated responses in mounting adaptive anti-TB immunity. In particular, studies in mice and in humans indicate that Th1-type CD4 T cells producing IFN-γ and cytotoxic CD8 T cells are important for controlling *M. tuberculosis* infection [3,4,5,6,7,8,9,10].

Humoral immunity mediated by B lymphocytes and antibodies has long been considered not to play a role in immune responses to *M. tuberculosis,* and controversial, yet very limited, results have been reported in human and mice studies [11,12,13]. However, more recent studies indicate that B lymphocytes participate to protect immune responses against *M. tuberculosis* [14,15,16]. Moreover, antibodies are produced against a variety of *M. tuberculosis* antigens in human and animal models, and total and mycobacterial-specific antibody glycosylation patterns correlate with the effective control of *M. tuberculosis* infection and may distinguish patients with active TB disease from subjects with latent TB infection (LTBI) [11,17]. Finally, a recent study in rhesus macaques found that following intravenous BCG vaccination, IgM antibody levels in the plasma and bronchoalveolar lavage were the best indicators of protection against *M. tuberculosis* challenge [18].

It has already been demonstrated that during TB the various hematopoietic populations are differently modulated based on the clinical stage of infection/disease [19,20,21]. However, studies aiming at correlating the frequency of circulating B lymphocytes with protective or pathogenic immune responses to *M. tuberculosis* have often yielded contradictory results, reporting either decreased or increased, or even unchanged, B cell percentages in patients with active TB disease, as compared to LTBI individuals and healthy subjects [22,23,24,25,26,27]. Finally, to our knowledge, there is no evidence in the literature that modifications of B cell percentages have an impact on antibody repertoire, including preexisting antibodies, to common microbial pathogens, given that residual antibodies may persist despite pharmacologic depletion of B cells in humans.

In this study, we analyzed the frequency, absolute counts, and phenotypic and functional subsets of B lymphocytes in the circulation of patients with active TB and subjects with LTBI and compared them to values in healthy donors. Furthermore, we performed a quantitative and qualitative analysis of the antibody repertoire, measuring serum levels of total Ig and their IgA, IgM and IgG isotypes as well as the titers of preexisting antibodies against a pool of common viral pathogens.

## 2. Materials and Methods

### 2.1. Characteristics of the Enrolled Individuals

For B cell and antibody analysis, a total of 61 individuals were prospectively enrolled, divided into 20 LTBI subjects, 25 active TB patients (active TB), and 16 healthy donors (HD) recruited from the University Hospital, Palermo, Italy (Table 1).

According to WHO guidelines, the diagnosis of active TB was assessed on clinical, microscopy, microbiological, biomolecular (GeneXpert, Sunnyvale, CA, United States), and radiological findings [2]. LTBI subjects were defined by a positive result to IGRA testing without any clinical finding or microbiological, biomolecular, and microscopy positive test. None of the TB patients had evidence of HIV infection or were being treated with steroids or other immunosuppressive or anti-tubercular drugs at the time of their sampling. The study was approved by the Ethical Committee of the University Hospital in Palermo, approval number 13/2013.

### 2.2. Cell Staining, Flow Cytometry, and Quantitative Analysis

Peripheral blood was drawn from LTBI subjects, active TB patients, and HD subjects. To assess the frequency of B cells, after Ficoll-Hypaque gradient separation, freshly peripheral blood mononucleate cells (PBMC) were surface stained, using mAbs to CD19, CD27, IgD (Miltenyi Biotec, Bergisch Gladbach, Germany). To evaluate the capability of B cells to produce GM-CSF or IL10, 1 × 10^6^/mL PBMCs were stimulated overnight in a 24-well plate at 37 °C, 5% CO_2_ in RPMI 1640 complete medium, with a final concentration of 1 μg/mL of Ionomycin (Sigma, St. Louis, MO, USA) and 150 ng/mL final concentration of phorbol myristate acetate (PMA) (Sigma). Non-stimulated cells were cultured in complete RPMI1640 as a negative control. Brefeldin-A (10 µg/mL, eBioscence, Thermo Fisher Scientific MA, USA) was added after 1 h of culture to block the Golgi activity. After stimulation, cells were harvested and stained, first with live/dead (L/D) marker (Zombie dye, Biolegend San Diego, CA, USA), then with mAb anti-human CD19 PerCP clone REA675, mAb anti-human CD27 PE-Vio^®^ 770 clone REA499, and mAb anti-human IgD FITC clone IgD26 (Miltenyi Biotec Bergisch Gladbach Germany). After surface staining, cells were fixed, permeabilized, and stained at room temperature for 30 min with mAb to anti-human GM-CSF PE clone DAVKAT (eBioscence) and mAb anti-human IL-10 APC clone REA842 (Miltenyi Biotec, Bergisch Gladbach Germany). Samples were acquired on a FACSARIA II flow cytometer (BD Bioscience San Jose, CA, USA) and analyzed using FlowJo v10 (BD Bioscience San Jose CA, USA). Flow cytometry acquisition of stained PBMC was performed using FACSAriaII cytometer (BD Biosciences San Jose, CA, USA), and for a clear analysis, at least 100,000 total events were acquired. Samples were analyzed by FlowJo software (Treestar Inc Ashland, OR, USA). Of the 55 samples, only 46 had sufficient viable cells to be included for B cell analysis (16 HD, 12 LTBI, 18 active TB). B lymphocytes were gated first by forward angle and side scatter profiles, followed by gating on singlets and viable cells expressing CD19. The analysis of each specific subset was performed using the gating strategy shown in Appendix A in compliance with the Minimal Information About T Cell Assays (MIATA) guidelines [8] and B-cell subsets defined based on the hierarchical model of human B-cell differentiation [9].

Full blood counts (FBCs) of peripheral blood collected in ethylene-diamine tetra-acetic acid (EDTA) containing tubes were performed by one clinical diagnostic laboratory using a five-part differential hematology analyzer (Beckman Coulter 4.500, Brea, CA, United States). Full blood count measurement was in accordance with strict quality procedures, including twice-daily high and low internal quality control, fortnightly quality controls done by the clinical laboratory QC scheme, and annual quality assurance as part of clinical laboratory QC scheme. The laboratory is accredited by the Italian National Accreditation System in accordance with international standards ISO17025/2005 and ISO 15189/2007. The absolute B cell count was performed by multiplying the total number of lymphocytes by the % of CD19^+^ cells and dividing by 100 [28].

Standardization of analysis and comparability of results were related to the use of reagents from the same producing company.

### 2.3. Dimensional Reduction and Exploratory Analysis

We performed unsupervised clustering on the two stimulated and control samples using FlowCT [29], an R-based package, to study the cell subpopulation in depth. Because this methodology fits better with homogeneous data, we limited it to 16 healthy controls, 10 LTBI subjects, and 14 active TB patients, and a semi-automated clustering and predictive modeling tool was utilized to analyze those FCS files concurrently. After pre-processing and quality control of previously FCS files (Appendix A), data were normalized and clustered using self-organized maps (FlowSOM). The dimensionality reduction technique uniform manifold approximation and projection (UMAP) was used to describe cell clusters. Heatmap clustering was used to annotate cell clusters, using median fluorescence intensity (MFI) values for each FCS.

### 2.4. Quantitative and Qualitative Analysis of Serum Immunoglobulins

Total antibody titers of IgA, IgM, and IgG from LTBI subjects (*n* = 18), active TB patients (*n* = 25), and healthy control individuals (HD, *n* = 10) were analyzed in whole serum by ELISA assay, and further analysis of the specific preexisting IgG levels against a pool of eight common pathogens (EBV, HSV-1/2, VZV, CMV, MV, RV, MuV, PV) was performed by Liaison^®^ DiaSorin chemiluminescence technology (DiaSorin, Saluggia, Italy) for the same three groups.

### 2.5. Differential Gene Expression Analysis and Pathway Analysis

Previously published PBMC expression profiling by array, including 9 active TB patients, 6 LTBI subjects, and 6 HD was downloaded from the Gene Expression Omnibus database (accession number: GSE54992). The gene expression microarray datasets were generated using Affymetrix Human Genome U133 Plus 2.0 Array, and the datasets were quantile normalized across arrays and transformed into a logarithmic scale. The background adjustment was performed using RMA, and PM-only was used as the PM adjustment method. Other packages such as pheatmap, ggplot2, ggrepel, and RColorBrewer were used for data processing and visualization. The Affy R package was used for differential gene expression analysis, and differentially expressed genes were visualized using the Enhanced Volcano package. The Log_2_ fold change (FC) threshold was set at 0.5 and −0.5, and the Benjamini-Hochberg adjusted *p*-value was used at a cut-off at 0.05. The transcript identifiers were mapped to Human Genome Organisation (HUGO) and GPL570 gene symbols, which were used for annotation.

In addition to genome-wide RNA expression analysis, Gene Set Enrichment Analysis (GSEA) was used to interpret gene expression data to yield insights into genes in B cell-related GO terms, including “B cell activation”, “B cell differentiation”, “B cell mediated immunity” and “B cell homeostasis”. Normalized gene expression matrixes from different study levels were then used for GSEA analysis using GSEA 4.1.0 software [30]. The Affymetrix Human Genome U133 Plus 2.0 Array chip platform was used for probe annotation. Adj *p*-value < 0.01 and FWER *p*-value ≤ 0.05 was considered as a significant enriched core.

### 2.6. Statistics

The median or mean was used for descriptive statistics for each parameter. The non-parametric Kruskal–Wallis was performed to determine statistical differences in the distribution of the results. Values of *p* < 0.05 were considered significant. The relationship between variables was evaluated using the Spearman rank correlation test. A two-side *p* < 0.05 was considered statistically significant. Data were analyzed using GraphPad prism, version 9.0 (GraphPad Software, San Diego, CA, USA). Standardization of analysis and comparability of results were related to using reagents from the same producing company.

## 3. Results

### 3.1. Circulating B lymphocyte Profiles during M. tuberculosis Infection

Either percentages or absolute numbers of circulating B lymphocytes were within the normal reference values in patients with active TB and in LTBI subjects (not shown). However, as compared to age- and gender-matched normal individuals (HD), patients with active TB had lower percentages and absolute numbers of circulating B lymphocytes on average (median value 13.5), which were even lower in LTBI subjects (median value 7.71). Such differences did not attain statistical significance, indicating that there is no substantial relationship between B lymphocyte percentages and the clinical features among groups (Figure 1A,B).

Furthermore, the Spearman rank test showed that the reduced B lymphocyte percentages and absolute values in active TB patients and LTBI subjects did not correlate with the percentages and absolute total lymphocyte counts (data not shown), suggesting that variations of lymphocyte percentages and absolute counts do not contribute to the decreased B lymphocyte values.

It is known that distinct subsets of B lymphocytes can be identified in the peripheral blood, some of which become over- or underrepresented in various disease settings, including *M. tuberculosis* infection. We then sought to determine the distribution of various phenotypic B lymphocyte subsets in active TB patients (*n* = 17), LTBI subjects (*n* = 12), and HD (*n* = 16). Thus, circulating B cells were divided into 4 subsets based on CD27 and IgD expression: naive (IgD^+^ CD27^−^), IgM memory (IgD^+^ CD27^+^), switched memory (IgD^−^ CD27^+^), and atypical (IgD^−^ CD27^−^), and their relative frequencies were evaluated in all three tested groups. As shown in Figure 2, naive and atypical B cells constituted the majority of B lymphocyte subsets in HD as well as in active TB patients and LTBI subjects. The phenotypic distributions of circulating B lymphocytes in active TB patients and LTBI subjects was characterized by nonsignificant overrepresentation of atypical B cells and a reduction of switched memory B cells (Figure 2). The switched memory B cell subset was down-represented in active TB patients, and even more in LTBI subjects in whom difference with HD attained statistical significance (Figure 2).

To further compare more precisely the memory B cell compartment in different TB conditions, we used a semiautomated empowered workflow known as FlowCT [31] to analyze large data sets that included preprocessing, normalization, and multiple dimensionality reduction techniques. Automated clustering was performed using FlowSOM on all CD19^+^ live B cells, cluster annotation was achieved by visualizing expression degrees of each marker on uniform manifold approximation and projection, and B lymphocytes were subclustered using phenotyping by accelerated refined community partitioning. By this approach, we analyzed 24,000 CD19^+^ cells from HD (*n* = 16 samples), 14,283 CD19^+^ cells from LTBI individuals (*n* = 10 samples), and 22,500 CD19^+^ cells from active TB patients (*n* = 14 samples) and combined all samples into one integrated dataset. Unsupervised clustering was then performed on this combined dataset of *n* = 60,628 CD19^+^ cells, which revealed 6 conserved clusters (Figure 3A). Four clusters matched to their original B cell subsets previously identified by classic FACS analysis. In addition, we found two cell clusters that could not be definitively linked to previously described B cell phenotypes. One cluster was IgD negative and expressed low levels of CD27, suggesting it represents a differentiation step in between switched memory and atypical B cells. The other cluster expressed low levels of IgD but was CD27 negative and could not be assigned to any specific B cell differentiation phenotype; it most likely reflected a second B naive population or a recently activated B cell population positioning in between naive and atypical B cell phenotypes. We then analyzed the 6 CD19^+^ B cell clusters for their percentage differences between different groups. Importantly, while all populations were present in all tested individuals, LTBI subjects had high percentages of atypical B cell populations and consistently lower percentages of naive, memory, and switched-memory B cells (Figure 3B,C). Hence, these results confirm those obtained by standard flow cytometry analysis; in addition, they highlight a previously unforeseen and significant reduction of the naive B cell subset in LTBI subjects.

### 3.2. Effect of M. tuberculosis Infection on Pro-Inflammatory and Regulatory B Cell Subsets

B cell heterogeneity is further highlighted by distinct cytokine-defined B lymphocyte subsets, with IL-10 and GM-CSF expression identifying regulatory (B-reg) and pro-inflammatory (B-inf) B lymphocytes, respectively. Hence, we quantified frequencies of B-reg and B-inf cell subsets in freshly isolated PBMC after short-term stimulation with ionomycin and PMA in the presence of monensin, followed by co-staining for intracellular cytokines and surface CD19. Standard flow cytometry analysis showed that frequencies of B-inf cells were very low and not statistically different in all three tested groups, while the frequency of B-reg cells in LTBI subjects was lower than in active TB patients and HD, although differences did not attain statistical significance, likely due to the broad value distribution and/or the relatively low number of tested samples (Figure 4A).

Unsupervised clustering showed three CD19^+^ IL-10^+^ B-reg clusters corresponding to naive, unswitched, and switched memory B cells and four CD19^+^ GM-CSF^+^ B-inf clusters corresponding to naive, unswitched memory, switched memory, and atypical B lymphocytes (Figure 4B). Moreover, unsupervised clustering also confirmed the lower frequency of CD19^+^ IL-10^+^ B-reg cells in LTBI subjects as compared to HD and patients with active TB (Figure 4C).

### 3.3. Impact of M. tuberculosis Infection on Antibody Repertoire and Preexisting Antibody Titers

We then asked whether *M. tuberculosis* infection affected the capacity of plasma cells to produce memory-type antibodies, at first by measuring serum levels of total Ig and their IgG, IgA and IgM isotypes. As shown in Figure 5, there were no clear differences amongst tested groups in total Ig, IgG, IgA, and IgM serum levels as determined by quantitative ELISA, demonstrating that plasma cell capacity to secrete different Ig isotypes is preserved during different clinical forms of *M. tuberculosis* infection.

However, it may still be possible that *M. tuberculosis*, despite preserving the antibody production capacity of long-lived plasma cells, may restructure the antibody repertoire, as occurs for instance after measles virus infection [32,33]. Therefore, we checked whether *M. tuberculosis* infection modulates preexisting antibodies that offer protection from other most-known human pathogenic viruses. To this aim, we profiled memory IgG antibody repertoires in active TB patients and LTBI subjects, relative to those observed in HD.

As shown in Figure 6 there were only very limited significant differences between active TB patients, LTBI subjects and HD in preexisting IgG antibody titers to ten different antigens from eight common pathogenic viruses: Epstein-Barr virus (EBV), Herpes Simplex virus (HSV-1/2), Varicella Zoster virus (VZV), Cytomegalovirus (CMV), Measles virus (MV), Rubeo virus (RV), Mumps virus (MuV), and Parvovirus (PV), namely lower anti-MV titers in LTBI subjects and higher anti-HSV-1/2 titers in LTBI subjects and active TB patients, as compared to HD. These results thus indicate that either active or latent *M. tuberculosis* infection does not impair, both quantitatively and qualitatively, the humoral immune repertoire.

### 3.4. Transcriptional Profiles of Circulating B lymphocytes during M. tuberculosis Infection

To corroborate and extend the findings that the B cell and antibody compartments are relatively maintained during *M. tuberculosis* infection, we performed a gene expression analysis in PBMCs of patients with TB, LTBI, and HD, using whole-genome transcriptional microarrays (Gene Expression Omnibus: GSE54992). After normalization and log_2_ conversion for the raw data of gene chips (Figure 7A and Appendix A), volcano plots showed no significantly different expression of 19,487 genes between active TB patients, LTBI subjects, and HD (Figure 7B). Moreover, we developed a 42-gene expression that can characterize the B cell population, and we found that this B-cell signature was similar and not significantly different in PBMC from active TB patients, LTBI subjects, and HD (Figure 7C).

We then used Gene Set Enrichment Analysis (GSEA) to evaluate differences between active TB patients, LTBI subjects, and HD in biological processes enriched in B lymphocyte signature gene sets. To this aim, we selected four GO categories: “B cell activation”, “B cell differentiation”, “B cell mediated immunity”, and “B cell homeostasis”. As shown in Figure 7D, and in agreement with previous results, none of the selected gene sets were differentially represented in active TB patients as compared to LTBI subjects and HD.

## 4. Discussion

*M. tuberculosis* remains a challenging and threatening pathogen, affecting many people globally. Epidemiological studies indicate that the progression of the disease is strongly associated with the immunological competency of the host. Characterization of the immune response to *M. tuberculosis* has largely focused on cell-mediated immunity [34,35,36], but recent studies evidence that humoral immunity can impact substantially on host defense against pathogens. Taking a more comprehensive approach, encompassing both cell-mediated and B cell and humoral immunity, characterizing immune responses to *M. tuberculosis* will likely gain new insights that can help to design anti-tuberculosis strategies, including immunotherapies and vaccines.

It is becoming clear that B cells and immunoglobulins contribute remarkably to shaping the immune response to generate protection against intracellular pathogens [37,38,39,40]. It has been seen in some studies that B lymphocytes, thanks to their ability to present antigens and produce antibodies and cytokines, can influence the population of T lymphocytes, essential in the defense against intracellular pathogens [41,42]. On the other hand, T lymphocytes, in particular CD4^+^ T lymphocytes, after exposure to pathogens, modify the responses of B lymphocytes, influencing the maturation of antibody affinity and class switch, the development of memory B cells and plasma cells, as well as cytokine production [43,44,45]. Therefore, characterizing the mechanisms by which B lymphocytes and humoral immune responses interact reciprocally with cellular immunity during tuberculous infection, may provide important information on new possible biomarkers to be used to discriminate the various stages of the disease or to design new antimicrobial strategies [46].

Results reported in this paper show that patients with active TB and LTBI subjects have slightly lower percentages and absolute numbers of circulating B lymphocytes as compared to healthy donors, but differences were not statistically significant. This finding was supported by gene expression and GSEA analysis, showing similar representation of B cell genes in active TB patients as compared to LTBI subjects and healthy donors. Thus, our results agree with previous studies reporting unaltered B cell frequencies in the blood of active TB patients and LTBI individuals as compared to healthy controls [47]. Conversely, other studies have shown significantly decreased or even increased [24,47] B cell frequencies in the peripheral blood of active TB patients [23,27,47,48] and LTBI [47] individuals compared to healthy controls. These conflicting data likely reflect differences between study designs and groups of patients enrolled, varying by age, gender, ethnicity, form, and severity of TB disease [13]. Accordingly, a very recent study of 217 cases of pulmonary TB detected low B lymphocyte counts only in patients aged over 60 years and with severe disease [49]. Analysis of B cell subsets by FlowCT and unsupervised clusterization revealed the presence of two more subsets in addition to the classic four subsets (naive, unswitched memory, switched memory, and atypical) distinguished by FACS analysis and based on CD27 and IgD expression. Despite these, six CD19^+^ B cell clusters were detectable in all tested individuals; LTBI subjects had high percentages of atypical B cell population and lower percentages of naive and switched memory B cells. A similar increased representation of atypical B cells in active TB patients and LTBI individuals, as compared to HD, has been documented in another study [27].

Differently from previous studies showing an increase of IL-10^+^ B-reg cells in active TB patients [26], we did not observe any differences in IL-10^+^ B-regs between active TB patients and HD. Conversely, LTBI subjects had lower percentages of IL-10^+^ B-regs compared to HD and active TB patients, but differences were not statistically significant. We also assessed the pro-inflammatory GM-CSF^+^ B-inf population, and we did not detect significant changes between active TB patients, LTBI individuals, or HD.

Finally, we assessed the impact of *M. tuberculosis* infection on the capacity of long-lived plasma cells to produce memory-type antibodies. The data herein reported show that serum levels of total Ig and IgG, IgA, and IgM isotypes were similar in active TB patients, LTBI subjects, and HD, demonstrating that plasma cell capacity to secrete different Ig isotypes is preserved during different clinical forms of *M. tuberculosis* infection. Similarly, there were no differences between active TB patients, LTBI subjects, and HD in preexisting IgG antibody titers to ten different antigens from eight common pathogenic viruses, clearly demonstrating that either active or latent *M. tuberculosis* infection preserves the antibody production capacity of long-lived plasma cells.

We would like to point out that our study has limitations due to the small sample size. Nevertheless, taken together, our results support the view that *M. tuberculosis* infection has very limited, if any, impact on the B cell compartment either in individuals who control infection (LTBI subjects) or in those who fail and develop active TB disease, and strongly support the view that *M. tuberculosis* does not significantly impair, either quantitatively or qualitatively, the humoral immune repertoire.

## Figures and Tables

**Figure 1 cells-11-02906-f001:**
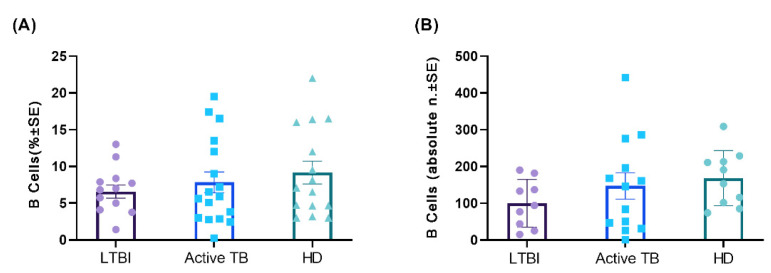
Quantitative analysis based on the frequency of CD19^+^ B cells in PBMCs (**A**) and absolute CD19^+^ B cell count (**B**) from peripheral blood among the groups; each dot represents one individual subject out of 13 LTBI, 17 Active TB, and 15 HD. Each horizontal bar represents the mean and the SE of each group.

**Figure 2 cells-11-02906-f002:**
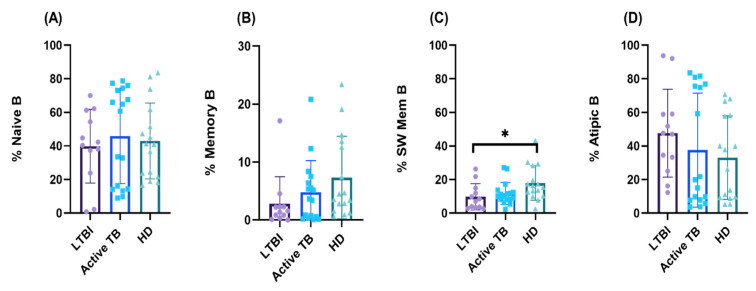
Phenotypic distribution frequency of Naive (**A**), Memory (**B**), Switched memory (**C**), and Atypic (**D**) B cells based on the expression of CD19, CD27, and IgD among the groups. We tested 12 LTBI, 17 Active TB, and 16 HD, as shown in the figure. The histograms represent the mean and SE of each group. Significance of differences between groups was compared using the Kruskal–Wallis test, * *p* < 0.05.

**Figure 3 cells-11-02906-f003:**
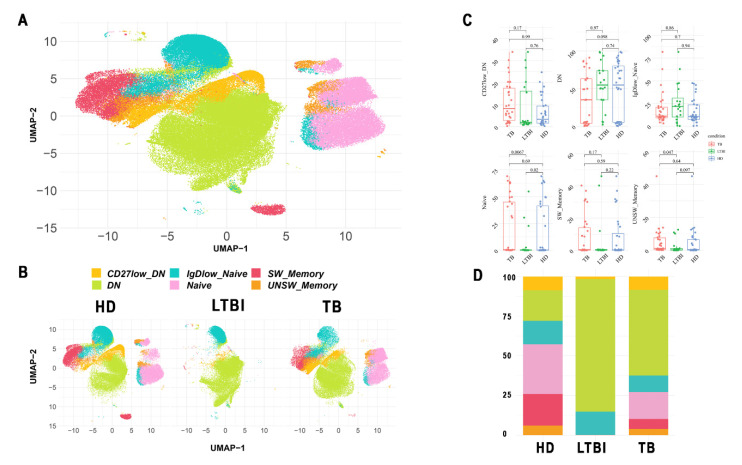
Clustering and cell subset identification: (**A**) After normalization and dimensional reduction, all B cells were clustered into six sub-clusters. An initial clustering step to sort all B cells was used. (**B**) The final clustering of different subsets clearly showed that no batch effect related to condition could be identified based on cell markers; the initial cell subset clustering using FlowSOM was performed on all cells. (**C**) B cell clusters identified by FlowCT evaluated according to condition; the two-tailed Anova test was used for sub-cluster changes in different conditions; *p*-value < 0.05 is considered a statistically significant change. (**D**) Proportion of each sub-cluster in different conditions.

**Figure 4 cells-11-02906-f004:**
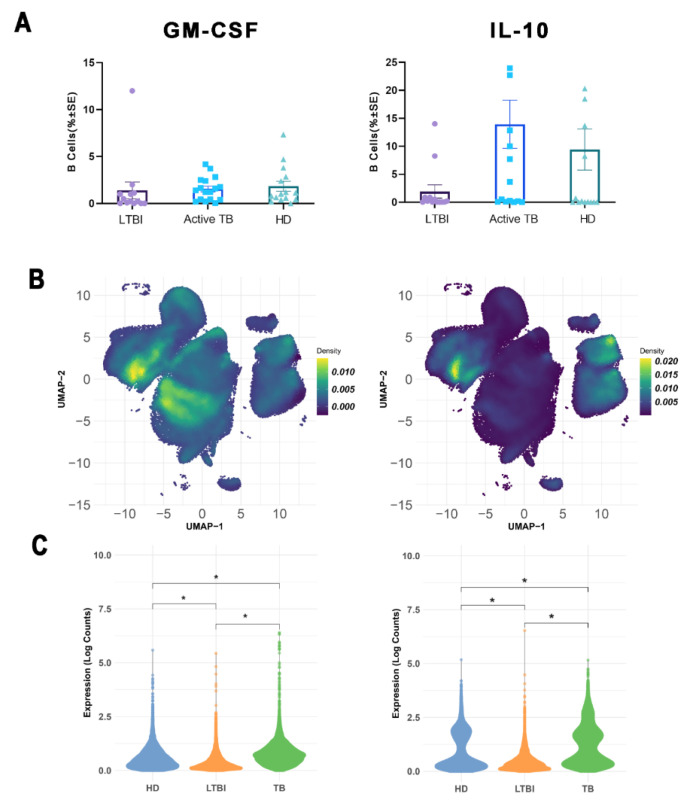
Analysis of GMCSF and IL-10 expression by CD19^+^ cells among the groups. (**A**) Analysis of GM-CSF (left) and IL10 (right) expression by CD19^+^ cells among the groups, after Ionomycin and PMA stimulation; each dot represents one individual subject out of 13 LTBI, 17 Active TB, and 15 HD. The scatter dot plots represent the mean and SE of each group. (**B**) UMAP density plots of B cells in stimulation conditions indicate cytokine expression in each relative cluster, dark blue shows no production of GMCSF and IL10, and yellow shows production of GMCSF and IL10. (**C**) Kruskal–Wallis one-way analysis of log_10_ counts of GMCSF and IL10 between different conditions in total B cells, * *p* < 0.05.

**Figure 5 cells-11-02906-f005:**
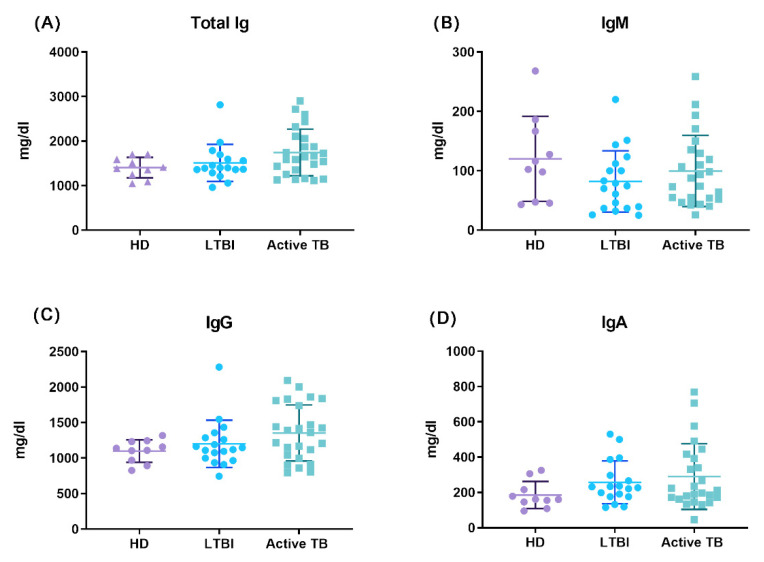
Quantitative analysis of total serum Ig (**A**), IgM (**B**), IgG (**C**), and IgA (**D**) immunoglobulins, from whole blood serum, among the groups; each dot represents one individual subject out of 20 LTBI, 25 Active TB, and 10 HD. The dot plots represent the mean and SE of each group.

**Figure 6 cells-11-02906-f006:**
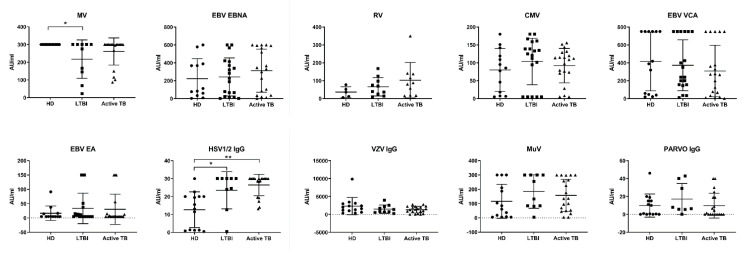
Quantitative analysis of preexisting IgG antibody titers to ten different common pathogenic viruses (MV, EBNA, RV, CMV, VCA, EA, HSV, VZV, MuV, PV), performed by ELISA assay; each dot represents one individual subject out of 20 LTBI, 20 Active TB, and 15 HD. The bars represent the mean and SD of each group. The significance of differences between groups was compared using the Kruskal–Wallis test; * *p* < 0.05, ** *p* < 0.01.

**Figure 7 cells-11-02906-f007:**
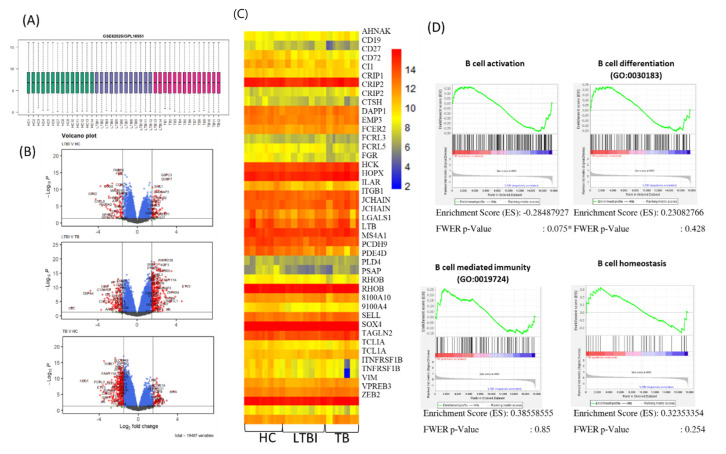
Gene expression and GSEA analysis revealed no significant change related to B cell signature. (**A**) Pre-processing and normalization of DNA microarray data shows the distribution of microarray after normalization and explains the uniform distribution obtained after implementing normalization, i.e., removal of noise from data. (**B**) Volcano plots showing genes significantly upregulated and downregulated in different groups. Blue dots: log_2_ fold change > 0, *p* < 0.01. Red dots: log_2_ fold change > 2, *p* < 0.05. The black line indicates adjusted *p*-value threshold (≤0.05) to set statistically significant results. (**C**) Affymetrix microarray data (heatmap) reveal no significantly differentially expressed B cell signature genes in different TB groups compared to healthy controls. (**D**) GSEA on gene set using GO terms shows only significant enrichment in B cell activation in TB groups versus HD. FWER *p* value < 0.01.

**Table 1 cells-11-02906-t001:** Characteristics of study groups.

	*HD*	*LTBI*	*Active TB*
Subjects	16	20	25
Male (%)	9 *(56)*	12 *(60)*	14 *(56)*
Mean Age (±*SD*)	34 *(±12)*	47 *(±14)*	52 *(±18)*
Origin			
Italy (%)	14 *(87)*	14 *(70)*	12 *(48)*
Eastern Europe (%)	2 *(13)*		5 *(20)*
Asia (%)		2 *(10)*	1 *(4)*
Africa (%)		4 *(20)*	7 *(28)*
QFT-IT pos. (%)	0 *(0)*	20 *(100)*	25 *(100)*
Microbiological pos. (%)			15 *(75)*
ZN pos. (%)			15 *(75)*
Molecular pos. (%)			20 *(80)*
Tuberculosis localization			
Lung (%)			22 *(88)*
Lymph node (%)			2 *(8)*
Disseminated (%)			1 (4)

QFT-IT = QuantiFERON-TB GOLD In-tube. ZN = Ziehl-Neelsen staining. HD = Healthy donors. LTBI = Latent TB Infection.

## Data Availability

The data presented in this study are available on request from the corresponding author.

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
