# Peer review of "Impact of Mycobacterium tuberculosis Infection on Human B Cell Compartment and Antibody Responses"

_cells, 2022, doi:10.3390/cells11182906_

Round 1

Reviewer 1 Report

The article by Manna et al “Impact of Mycobacterium tuberculosis infection on human B cell compartment and antibody responses”  compares the humoral response in healthy controls, and patients with either latent tuberculosis or active tuberculosis. This is a descriptive study that is very well written. The authors conclude that the B-cell compartment is not significantly altered in either form of TB infection. The conclusions follow from the data presented which are generated from the appropriate laboratory techniques. This work adds to a growing body of literature looking at the different parts of the immune response in the setting of TB infection.

There are some opportunities to improve the manuscript:

1.     The population description is listed as Table 1 in the manuscript then later found as a supplementary table. This information is important enough that it should be placed in the manuscript so the reader can assess the population characteristics directly when reviewing the data. It would also be helpful to know how far along these patients were in the treatment for TB just initiating therapy, towards completion as this could affect the results.  

2.     Page 4 line 150- please clarify whether the 24 patients were active TB or LTBI.

3.     Page 4 line 159-160 in general throughout the manuscript when fewer samples were used some explanation as to why not all samples were included or how the samples to be included were chosen would be useful. Also a statement that the characteristics of the patients who provided those samples were  statistically similar to the available population with respect to age etc. to ensure there is not a selection bias.

4.     Page 5 line 221-222 Is this the same number of patients as Figure 1. Would be helpful to include in the manuscript.

5.     Page 6 figure 2. It is not clear what is meant by “Each column 235 represents one group out of 12 LTBI, 17 Active TB, 16 HD”. What is the group? For consistency and clarity consider showing Figure 2 the same way as other Figures such as Figure 1, showing the data point for each subject.

6.     Page 6 line 259.  “Importantly, despite all populations could be found in all tested individuals,”

Consider changing, “…while all populations were present in all tested individuals…..

7.     Page 9 line 322 viruses need abbreviations spelled out. Also for Figure 6 when looking at MV titers, did they control for vaccination status and time since vaccination.

8.     Figure 7 is generally of poor quality and blurry and the figure legend for 7D is incomplete and 7D cannot be read

9.     Discussion line- 370 and other places (line 415), the authors mention recent or other studies but provide no references.

10.  In many places the discussion is a recapitulation of the results that can be truncated. The authors offer differences in the patient population as a reason for some of these results being different but do not describe what is different about this patient population. Also given this is looking at the B-cell compartment, were the patients with TB in the lymph nodes outliers or excluded from any of the analysis. In general how this patient population differs from other studies should be discussed in more detail. Also the small sample size should be mentioned as a limitation.

Author Response

We would like to thank the reviewers for the very positive comments to our manuscript.

Reviewer 1

  1. Accepting the reviewer’s recommendation, we have placed Table 1 with patients characteristics in the manuscript. As stated in Section 2.1, none of the TB patients were treated with anti-tubercular or other drugs at the time of sampling.
  2. Unsupervised clustering was performed on samples from 16 healthy donors, 10 LTBI subjects and 14 active TB patients. We apologize for the mistake which has now been corrected both in Materials and Methods and in Results sections.
  3. In some experiments we used fewer samples because lower cell numbers or serum amounts were recovered, or because high mortality rates after Ficoll-Hypaque PBMC isolation or in vitro In any case, in each experiment the tested samples fully reflected the characteristics of the whole population, such excluding any selection bias. We have now placed a general statement in Materials and Methods, Section 2.1.  The reviewer’s suggestion is accepted with thanks.
  4. Accepting the reviewer’s suggestion, we have included also in the manuscript the number of patients as in Figure 1.
  5. Accepting the reviewer’s suggestion, we have modified Figure 2 to be shown in the same way as Figure 1.
  6. The statement on page 6, line 259 has been changed as suggested by the reviewer.
  7. Viruses abbreviations have been spelled out as recommended by this reviewer. Concerning the vaccination status, all Italian patients stated they received two doses MMR (measles, mumps, rubella) vaccine, one at around 9 months and a booster dose at around 18 months. We do not have information on the vaccination state of the african, asian and the one eastern Europe TB patients.
  8. We apologize for the very poor presentation of Figure 7. We now provide a better quality Figure 7 and a corrected legend.
  9. We have provided missing references to other quoted studies in the Discussion section.
  10. Accepting the reviewer’s suggestion, we have eliminated the repetitive parts of the Discussion. As per the reviewer’s question, the results in two patients with lymphatic TB and one patient with disseminated TB were not outlier and were included in the study.  Finally, we also mentioned the small sample size of our study as another possible limitation.

Reviewer 2 Report

The authors do a good analysis to re-investigate the role of B-Cells during latent and active TB infection and find outcomes to what is previously known and demonstrated. However, one area that requires further attention in the field is to evaluate the role B cell place in modulating the function of cells within the granuloma such as monocytes and T cells. However, this seems beyond the scope of this study.

PMA/ Ionomysin restimulations are often very non-specific, especially from human samples as they are likely to have encountered several antigens during their lifetime. Can the experiments be repeated with a more specific TB peptide or heat inactivated TB to record MTB specific B cell responses.

Author Response

We would like to thank the reviewers for the very positive comments to our manuscript.

Reviewer 2

  1. We fully agree with the reviewer’s point that looking at the B cell compartment in the context of the granuloma, but as mentioned by the reviewer, this issue was beyond the scope of our study.
  2. We fully agree that PMA/ionomycin stimulation is not specific. However, this is the protocol used to assess B-reg and B-inf subset and has been used to asses the impact of M. tuberculosis infection on the whole B cell repertoire. recommendation, we have placed Table 1 with patients characteristics in the manuscript.  As stated in Section 2.1, none of the TB patients